# Addressing the psychological impact of climate-induced disasters on young people in Africa: Challenges and pathways forward

Usoro Udousoro Akpan[1] ⓘ, Ibrahim Khalil Ja'Afar[1] ⓘ and Sinclair Chidera Eke[2] ⓘ

[1]Warwick Medical School, University of Warwick, Coventry, UK and [2]Department of Public Health, University of Derby, Derby, UK

## Perspective

climate change; disaster; mental health; young people; community engagement; Africa

**Corresponding author:**
Usoro Udousoro Akpan;
Email: usoro.akpan@warwick.ac.uk

## Abstract

Climate change is exacerbating the frequency and severity of disasters across Africa, with profound psychological consequences for young people. This paper examines the mental health impacts of climate-related events like droughts, floods and extreme weather on African youth. It explores how climate stresses compound existing societal issues, affecting young people's well-being. Studies highlighted indicate events strongly associated with negative emotions, anxiety, post-traumatic stress disorder (PTSD) and depression among youth. Vulnerabilities are due to disrupted community contexts and limited support systems. Challenges in providing adequate care are also reviewed, with African health systems grappling with a shortage of professionals and inadequate youth-focused care. This article proposes solutions centred on integrated disaster response, community resilience programmes and specialised youth services. Recommendations involve prioritising mental health education, establishing accessible services and collaborating with local partners. The overall aim is to comprehensively address African youth mental health needs in climate-changing contexts through holistic multi-stakeholder approaches, building coping skills and promoting wellness.

## Impact statement

Climate change is increasingly exacerbating the psychological suffering of young Africans exposed to climate-related disasters like extreme weather events, droughts and floods. The chronic stress induced by these environmental shocks, compounded by uncertainty about the future, has far-reaching negative consequences for their mental health and well-being. Unless urgent action is taken, Africa risks raising a damaged generation psychologically scarred by the impacts of the climate crisis. It is therefore imperative to prioritise youth mental health support in climate policy and programming on the continent. Integrating psychosocial care into disaster response as well as fostering community resilience through culturally-sensitive awareness initiatives can help build coping skills to strengthen young minds against climate threats. Moreover, training more mental health practitioners and establishing youth-friendly services accessible even in remote areas are desperately needed. By addressing the root causes of eco-anxiety and stress among African youth, we can help them overcome trauma, stay hopeful and realise their full potential to lead their communities and continent despite environmental hazards. A healthier generation psychologically equipped to face climate realities is key to achieving long-term sustainability, stability and prosperity across Africa. Failure to act on the profound psychological toll of climate upheaval on young lives would be denying them, and the future of the continent, a fair chance at wellness. Urgent multi-pronged action is thus warranted to shield African youth's mental health from current and impending climate impacts.

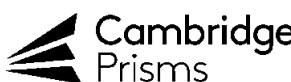



## Introduction

Climate change and weather-related disasters not only result in direct impacts such as destruction and trauma but they also have indirect consequences like strained resources and interrupted community and societal functioning (Hickman et al., 2021). These lead to climate-induced anxiety among young people, thus affecting their psychological and emotional well-being (Ramadan et al., 2023). The experiences faced by young people in Africa today are increasingly marked by climate-induced disasters, resource shortages, inequality, and refugee crises, which can reduce social cohesion and also increase political instability (Tanner et al., 2022). A study by Gasparri et al. (2022) indicates that climate change has a profound impact on the psychological and emotional well-being of young people, with climate-related events leading to negative emotions and mental ill-health. The psychological impact of climate-related disasters disproportionately affects children, adolescents, and young people, making them more susceptible to

conditions like PTSD, eco-anxiety, and depression, with long-lasting effects (Gasparri et al., 2022). As climate-induced disasters escalate, there is a growing need to understand their impact on children and young people, especially in the face of increasing climate threats (Gao et al., 2023). Mental health services in Africa must collaborate with young people to develop strategies that enhance resilience to the psychological impacts of climate change (Godden et al., 2021). Addressing the psychological impact of climate-induced disasters on young people in Africa requires a holistic approach that considers the unique vulnerabilities and challenges faced by them. By understanding the psychological effects of climate change and implementing targeted interventions, it is possible to support the mental health and well-being of young people in Africa in the face of environmental challenges.

### Climate-induced disasters, psychological effects on young people in Africa and the challenges in providing mental health support

Climate-induced disasters in Africa pose a significant challenge due to the continent's vulnerability to the impacts of climate change. Several studies have pointed out the increased frequency and severity of disasters such as windstorms, floods, droughts and extreme weather events in various regions of Africa (Tirivangasi, 2018; Shokane, 2019). Southern Africa, in particular, has historically been prone to climate-induced disasters like droughts and floods, which have had repercussions on the well-being of communities (Kamara et al., 2018). The consequences of these disasters go beyond immediate losses, impacting economic growth, agriculture, food security, and even contributing to conflicts in the region (Shimada, 2022). Moreover, the impact of climate change on mental health in Africa is a growing concern. A study by Atwoli et al. (2022) has shown that climate change-related disasters have adverse effects on mental health, such as trauma, anxiety disorders, depression and suicidal thoughts. This also necessitates more research and awareness to prevent potential mental health crises in the future. Environmental shocks resulting from the climate crisis not only affect physical health but also have severe implications for mental well-being (Atwoli et al., 2022).

The psychological effects on young people in Africa exposed to climate disasters are profound and multifaceted. A study by Tosam and Mbih (2015) highlighted the association between climate change awareness, government inaction, and negative psychological outcomes among young people in Africa. The psychological impacts of global climate change, including anxiety and distress, further exacerbate the mental health challenges faced by young people in the context of climate disasters (Atwoli et al., 2022). Extreme weather events resulting from climate change have been shown to affect the mode and behaviour of young people and potentially lead to long-term psychological effects (Mutua et al., 2023). Also, concerns about climate change have been linked to distress in the general population, with certain individuals being more vulnerable to psychological distress (Atwoli et al., 2022). Schwartz et al. (2023) studied climate change anxiety and its impact on mental health and identified environmental activism as a potential buffer against climate-related psychological distress. The heavy psychological burden imposed on young people by climate-related disasters and the anticipation of future challenges has been recognised, thus necessitating interventions to support their mental well-being (Mugeere et al., 2021).

The challenges in providing mental health support to young people in Africa exposed to climate disasters are multifaceted and require tailored interventions. Africa faces a critical shortage of mental health professionals, thereby exacerbating the inadequacy of mental health services for young individuals with common mental health disorders (Mutero et al., 2022). Limited knowledge, which has resulted in misdiagnosis, stigma, and insufficient interventions for addressing mental health issues among young people in Africa, further compounds the challenge (Jumbe et al., 2022). The design of mental healthcare systems in many African countries is often inadequate to meet the unique developmental and cultural needs of young people, hindering access to appropriate support (Ndetei et al., 2022). Structural barriers such as practical challenges in accessing and engaging with mental health services, present significant obstacles for trauma-exposed young people seeking mental health care (Ellinghaus et al., 2021). Moreover, the prevalence of mental health disorders among young people is on the rise, necessitating urgent attention to address mental health needs in this population (Ravenna and Cleaver, 2016).

### Addressing the challenges to providing mental health support to young people impacted by climate-induced disasters

Integrating mental health support into disaster preparedness and response efforts is important. A study by James et al. (2020) showed that interventions teaching coping skills can enhance functioning, reduce hopelessness, and improve participants' ability to engage with disaster preparedness content. Additionally, an integrated approach to mental health and disaster preparedness, focusing on community participation and social support, has been linked to improved mental health outcomes in adults, which can also have the same impact on young people (James et al., 2020). So, addressing the psychological impacts of climate change and disasters involves fostering resilience among young people. This can be achieved through community-based interventions that promote social involvement and provide psychosocial support (Gislason et al., 2021). Involving young people in mental health literacy programmes and equipping them with coping strategies can enhance their resilience to climate-induced stressors (Elkington et al., 2011). Interventions should take into account the psychological effects of disasters on young people, such as anxiety, depression, and post-traumatic stress symptoms. Therefore, providing mental health services that address these specific needs, including reducing victimisation and enhancing community participation, can lead to improved mental health outcomes (Paceley et al., 2017). Additionally, interventions focusing on reducing distress, enhancing readiness for participation, and safeguarding privacy are crucial for engaging young people in mental health support programmes (Campbell et al., 2020). By implementing a combination of community-based interventions, mental health literacy programmes, and tailored mental health services, stakeholders can strive to overcome the challenges in providing mental health support to young people affected by climate-induced disasters in Africa. These efforts should aim to build resilience, address specific psychological impacts, and promote community engagement to support the mental well-being of young people facing climate-related challenges.

### Recommendations

To comprehensively address the mental health needs of young people in Africa impacted by climate-induced disasters, several recommendations can be proposed. First, it is crucial to prioritise the integration of mental health education and awareness

programmes into school curricula and community initiatives to enhance mental health literacy, promote social cohesion, and engage youth in mental health support initiatives to build resilience and enhance coping mechanisms. Additionally, establishing youth-friendly mental health services that are accessible, culturally sensitive, and tailored to the unique needs of young people can improve mental health outcomes. Utilising telehealth services in hard-to-reach communities will also be helpful. Collaborating with local communities, traditional healers, and youth organisations to destigmatise mental health issues and provide psychosocial support is essential. Investing in training programmes to increase the number of mental health professionals, particularly in underserved areas, and incorporating mental health components into disaster preparedness and response plans can strengthen the mental health support system for young people in Africa facing the psychological impacts of climate disasters. Furthermore, investing in the collection of local data on trauma, early screening and culturally appropriate assessment tools will further guide the development of multi-pronged interventions addressing key issues like anxiety, depression and post-traumatic distress. With concerted efforts to make services accessible even in remote areas through community health systems, the mental well-being of African children and adolescents faced with intensifying climate disruptions can be better protected.

## Conclusion

Addressing the challenges in providing mental healthcare to at-risk youth requires holistic, multi-pronged interventions. Promising avenues to build resilience and strengthen provision of care identified from recent studies include integrating psychosocial support into disaster management, community resilience programmes and specialised youth services. However, more research is needed to fully understand context-specific needs and effects across Africa. If left unaddressed, the mental health consequences of climate hazards on developing young minds risk undermining long-term well-being and stability in Africa. It is therefore imperative to prioritise youth-centric policy and programming responses. Mainstreaming mental health education, establishing accessible services, and collaborating closely with local partners hold potential to better shield African children and adolescents from the profound psychological toll of our changing climate. A healthier generation equipped to face environmental realities will be key to overcoming adversities and securing a prosperous, sustainable future for communities and the continent as climatic disruptions intensify. Robust, urgent and coordinated action across sectors is warranted.

**Open peer review.** To view the open peer review materials for this article, please visit http://doi.org/10.1017/gmh.2024.77.

**Data availability statement.** Data sharing does not apply to this article as no datasets were analysed.

**Author contribution.** This manuscript was conceptualised by U.U.A. I.K.J. and S.C.E. assisted in drafting the manuscript. All the authors read and approved the final manuscript and accepted responsibility for the decision to submit it for publication.

**Financial support.** This manuscript received no financial support from any funding agency in the public, commercial or not-for-profit sectors in the writing of this paper.

**Competing interest.** There is no conflict of interest to declare.

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
