## [Reviewer Report]

Congratulations for the well written manuscript. It would be better if you could include a subsection to highlight the limitations of the study.

---

## [Reviewer Report]

Thank you for the opportunity to review this interesting paper on an important topic.

My main concern with this paper is that I don’t believe it makes enough use of the existing body of evidence relating to climate change and mental health impacts in Africa, and the smaller body of work specifically focused on young people. Most of the citations used in this paper are drawn from the broader research base and not from Africa or other developing countries.

A very cursory search of the evidence base identified papers that draw very specifically on Africa, and which could have been included here – e.g.

https://jamanetwork.com/journals/jama/fullarticle/2810968

https://doi.org/10.1016/j.scitotenv.2023.163420

https://bmcpsychiatry.biomedcentral.com/articles/10.1186/s12888-024-05568-8

https://www.sciencedirect.com/science/article/pii/S2667278221000845

https://doi.org/10.1016/S2542-5196(23)00234-6

http://dx.doi.org/10.7196/samj.2019.v109i9.14327

As an example, I see that the current paper cites a 2022 review by Atwoli et al. which looks at the links between climate change and mental health in the broader African population rather than specifically focusing on young people. I note that Atwoli et al. draw much more on African-specific research, and provide more detail on potential pathways for climate concerns to impact mental health, as well as more detail on potential responses than have been provided in the current paper.

While I think the topic is worthy of publication, I do not think the current document is publishable because it does not synthesise the local evidence base well.

If the paper does get accepted for publication then I do have a number of tiny comments regarding the text in the PDF that was submitted for review:

Page 1

Line 8 - In the title you refer to young people - but here refer to children. It may be worthwhile defining your target population at some point as these terms are not always used interchangeably.

Line 30 – referring to a “lost” generation does not seem to be appropriate here - the generation wouldn’t be lost, but they may be damaged by lack of care.

Line 57 – The use of the Sanson et al. reference here is an example of my main comment above that the paper does not draw enough on African-specific research. This reference isn’t primarily about Africa, whereas the sentence is focused on Africa.

Page 2

Line 26 - These two citations should be merged into the one set of brackets

Line 32 – “have shown” should be “has shown”

Line 33 - The Atwoli paper is quite detailed and the findings could be drawn upon more here potential mental health outcomes.

Line 40 – Again, the section header and the previous sentence locate the focus in Africa, so the reader might take it that this reference by Hickman et al. reports on African findings - which it does not - nor do any of the other references cited in the rest of this paragraph

Line 60 - Again, could go into more detail here rather than referring to “Limited knowledge”

Page 3

Line 6 - Again, the sentence is talking about the African healthcare system but the McGorry paper does not.

Line 24 - The James paper is not focused on young people, so you shouldn’t generalise the findings to young people. Would be better to make it clear above that the James study was on adults and that might be the same for young people - or to find research specifically on young people.

Line 55 - Accessible mental health services is challenging across such a diverse and huge continent involving many different countries. The challenges in providing such services should be explored more - including the potential to do so using telehealth (which has its own challenges).

Line 57 - You are skipping between US and British English here and elsewhere - need to be consistent.

Page 4

Line 24 - Previously, you have hyphenated well-being but here you have dropped the hyphen - I suggest you be consistent.

Page 5

Line 39 – this is a preprint that has now been published (see https://journals.plos.org/plosone/article?id=10.1371/journal.pone.0281655) – so suggest you update your reference.

Page 7

Line 14 – again, this is a pre-print that has now been published (see https://onlinelibrary.wiley.com/doi/full/10.1111/eip.13374)

---

## [Reviewer Report]

Congratulations for this manuscript. The concept and methodology are sound. It would be great to highlight the limitations of the study.

---

## [Editor Report]

Discretionary comment regarding response to reviewer 1

"To reviewer 1:

I did not include limitations of the study because this is just a perspective and no primary data collection was done".

This response in its response could be a limitation of the paper, dont you think so?